# Exploring the Implications of a Needs-Based Pharmacy Education Framework Modelled on Population Health: Perspective from a Developing Country

**DOI:** 10.3390/pharmacy7030116

**Published:** 2019-08-14

**Authors:** Angeni Bheekie, Mea Van Huyssteen, Renier Coetzee

**Affiliations:** School of Pharmacy, University of the Western Cape, Bellville 7535, South Africa

**Keywords:** health systems, needs-based pharmacy education, logical framework, population health, relevance, social accountability, developing country

## Abstract

Globally, health education reform is directing efforts to strengthen the health system through collaboration between health education and health services. However, collaborative efforts vary between developed and developing countries as the health needs, economic constraints, and resource availability differs. In developing countries, resource allocation is weighed in favor of interventions that will benefit the majority of the population. The question that emerges is: How could health education, service, and research activities be (re-)aligned to optimize return on investment for the health system and society at large? This paper proposes a needs-based pharmacy educational approach by centralizing population health for a developing country like South Africa. Literature on systems-based approaches to health professional education reform and the global pharmacy education framework was reviewed. A needs-based pharmacy educational approach, the population health model which underpins health outcome measurements to gauge an educational institution’s effectiveness, was contextualized. An evaluation framework to determine the pharmacy school’s effectiveness in strengthening the health system could be applied. A needs-based pharmacy educational approach modeled on population health could: Integrate resources from education, service, and research activities; follow a monitoring and evaluation framework that tracks educational outcomes; and engage with external stakeholders in curricular development and assessment.

## 1. Background

The World Health Organisation (WHO) attests “the needs of the health system should shape the way in which the workforce is educated—not the other way around” [1]. To achieve this, interdependence between the health system and health education systems is crucial for attaining the goal of health for all [2]. However, the execution of collaborative efforts may vary between developed and developing countries as the health needs, economic constraints, and resource availability differs. In resource constrained environments decisions regarding resource allocation needs to be weighed meticulously in favor of interventions that will benefit the largest proportion of the population, often leaving very little resources for individual directed health interventions. This might imply a different relationship between patient-centered and population-based approaches of health education and health care in developed and developing countries.

Similarly, the International Pharmaceutical Federation (FIP) advocates for a needs-based approach to guide pharmacy education [3], and recognizes that a global competency framework using a single global curriculum would not adequately address the needs of developed and developing countries [4]. In this regard, the FIP’s global competency framework offers detail on patient-focused competencies with less emphasis on population-focused competencies [4]. In designing curriculums for a resource constrained context, the focus on patient-centered care is not optimal, since the deployment of resources should be strategic to cover a larger cohort of people, whereby directing resources at an inter-individual level would not have as much impact in improving population health indicators. The question this raises for pharmacy education institutions in developing countries is: How could their education, service, and research activities be adjusted to pool these resources with those of the health system to ensure the most effective return on investment for the health system and society at large? The design of a pharmacy education framework directed at population health is yet to be documented for a developing country. 

This paper articulates a needs-based pharmacy educational approach from a developing country perspective by centralizing population health as the primary reference point. Firstly, the global health perspective on professional education reform towards a systems-based approach is outlined. Secondly, the FIP’s global educational guidelines for pharmacy education are contextualized. Thirdly, we examine the status of pharmacy education and practice in a resource-limited developing country, using South Africa as an example. Finally, a needs-based pharmacy education framework for a resource-limited setting which underpins population health is discussed. 

## 2. Systems-Based Approach to Reforming Health Education

The Lancet Commission’s proposed educational framework aimed at addressing equity in health care advocates for an interdependent approach to reform all health professional education [2]. Such interdependence emerges from the supply and demand cycle that exists between the health education system, responsible for the supply of health care workers, and the demands of the health care system, which is determined by population health needs. Since the starting point of this interdependence rests upon serving the health needs of the population, the health of the population becomes a key determinant for the type of health education (i.e., professional competencies) that is offered to improve health care. In order to implement an interdependent educational approach, population health thus becomes the focal point, using evidence-based approaches to address inequity.

Kinding and Stoddart (2003) proposed that population health be “concerned with both the definition and measurement of health outcomes and the roles of [health] determinants” in defined populations [5]. The authors defined population health as “the health outcomes of a group of individuals, including the distribution of such outcomes within the group”. In addition, the population health determinants and outcomes are influenced by policies and interventions, which are not only confined to the health system, but permeate across multiple dimensions such as the environment, social circumstances, employment, and other role players. 

However, the historical disconnect between population health and the health system, illustrates that medical care, which was provided through the health system, was neglected as a determinant of population health [6]. Access to health care, as calculated in a developed environment, contributes about 10% to reducing an individual’s premature mortality [7] yet, in the achievement of health for all, getting on top of population health is deemed the next big challenge for developed countries’ health systems [8]. When health education is viewed through the lens of population health, which is based on determinants and outcomes, a logic model [9] offers a systematic framework to measure the effectiveness and quality of educational interventions that address health needs [10].

A value-based health care system “explicitly prioritises health outcomes that matter to patients relative to the cost of achieving those outcomes” [11], by optimizing the use of finite resources to achieve patient-centered care. “At the heart of such a model are the payment mechanisms that either encourage effective treatments that deliver value or create disincentives for those that are not cost effective and do not deliver value” [11]. The Economist Intelligent Unit (EIU, 2016), evaluated alignment among developed and developing countries with value-based health care components such as health coverage, access and resources, health professional education and training, patient outcomes data, bundled payments, and incentives. The EIU study findings revealed that lower-income and developing countries attained low alignment with value-based health care as increasing quality and access to basic health services is a priority of the health system. Countries with a total health expenditure of less than 5% of the gross domestic profit had very low overall health coverage for its population. Most developed countries displayed moderate alignment with value-based care approaches. Countries with higher health care spending greater than 10% of the gross domestic profit used electronic health records, operated on bundled payment system, received stakeholder support, and received cost cutting incentives in implementing value-based health care. Developed countries were better equipped to implement professional training in value-based patient-focused training than developing countries. Overall, what seems to be universally lacking is co-ordination of patient health data, and interoperable information technology whereby systems can exchange and interpret data for a user’s understanding to track outcomes and costs to determine how health systems are delivering value to individual patients. In this regard, the patient-centered, value-based health care system for a developing country might be less appropriate because of limited resources, as addressing the priority health needs of the majority of the underserved population, might be a more appropriate option.

## 3. Global Pharmacy Education Guidelines

Globally, in terms of needs-based pharmacy education, patient-centered drivers for change in the curriculum have been evident for both developed and developing countries [3]. In developed countries, this need is directed towards specialization areas in pharmacy, whereas, in developing countries this need is towards more patient-centered curricula in general, and public health pharmacy [3]. The patient-centered approach depends primarily on adequate human resources, because of its inter-individual nature, and is well suited for the specialist services needed in developed countries. As such, the FIP’s global competency framework provides detailed information on patient focused pharmaceutical care competencies [4]. Of the four competency domains in the FIP’s global competency framework, the least descriptive quarter focuses on population and public health. The population focus consists of pharmaceutical public health (health promotion, medicine information, and advice) while the system focus includes organization and management (budget and reimbursement, human resource management, improvement of service, procurement, supply chain, and workplace management). The patient focus relates to pharmaceutical care (patient consultation and diagnosis, and medicine-related functions (assessment, dispensing, compounding, monitoring)); and the practice focus is directed at professional or personal development (communication skills, continuing professional development, legal and regulatory practice, professional and ethical practice, quality assurance and research in the workplace, self-management). 

Indeed, a comparative study from four developed countries, which mapped the learning outcomes for pharmacy graduates against this framework, found that it was geared towards producing patient-oriented medicine experts, but in its reference to competencies lacked on key aspects such as interprofessional teamwork and leadership attributes [12]. However, a recent consensus guidance document is aimed at promoting quality interprofessional collaborative practice, with the primary goal to improve health outcomes [13].

## 4. Needs-Based Pharmacy Education: A South African Perspective

The FIP’s needs-based professional education model is applied and interpreted to a South African context, which is representative of a developing country in terms of resource constraints [14]. The model is discussed in relation to locally determined needs, the pharmaceutical services, and related competencies, concluding with pharmacy (higher) education. 

### 4.1. Needs

South Africa’s population is estimated at 56 million [13] consisting predominantly of black Africans (80%). South Africa has been rated the most unequal society in the world (a Gini coefficient of 0.7), with almost half (45%) of the population still living on approximately $2 per day and more than 10 million people living on less than $1 per day [15]. The quadruple burden of disease, which includes HIV/TB, maternal and child health, noncommunicable diseases, and violence and injuries, is skewed towards poor socioeconomic groups which primarily affect mothers and children. In addition, inequality in health service provision between the urban and rural, and rich and poor, are reflective of the country’s fragmented health system [16]. 

South Africa’s rights-based constitution endorses equitable health service delivery, which is clearly illustrated from two landmark constitutional court case rulings. In the first case, the government’s inability to offer long-term renal dialysis to a patient due to insufficient funds, led to the court ruling that the under-resourced hospital was not in a position to offer such a specialized and expensive service for the benefit of only one patient [17]. In the second case, access to the antiretroviral nevirapine which is used to prevent mother-to-child transmission of HIV, was restricted to certain health facilities, which were classified as the government’s study (‘pilot’) sites. In this case, the court ordered the government to extend availability of nevirapine to all health facilities as quickly as possible, partly because of the positive impact this would have for the survival of many infants [18]. The two cases clearly illustrate South Africa’s commitment to the constitution, by allocating its limited resources to benefit the majority of the population, aspiring towards equity in health care. 

However, recent health expenditure illustrates a gross inequality in terms of health care needs versus benefits. It is estimated that the richest 20% of the population, with a ‘health need share’ of less than 10%, receive 36% of total health care benefits, while the poorest 20% with a ‘health need share’ of more than 25%, receive only 12.5% of the benefits [18]. In addition, the auditor general’s report on the performance audit of pharmaceutical management nationally revealed several challenges to medicine availability, and a key policy and planning level finding was that the pharmaceutical budget was not always aligned to the actual health care needs of the uninsured (underserved) population [19]. In addition, an evaluation of the outcome of the Department of Health Strategic Plan (2010–2013) showed persistent challenges that included the complex quadruple disease burden, the poor quality of public health care, an ineffective and inefficient health system, and spiraling private health care costs [19].

The process of attaining universal health coverage, a target of sustainable development goal three [20], is underscored by South African policies such as the National Development Plan (NDP) 2030 [21] and National Health Insurance (NHI) [19]. These policies centralize primary health care (PHC) to reorientate resource allocation and service interventions towards population health. In terms of resource allocation, the PHC approach is directed at health promotion and disease prevention as a mechanism to address the social determinants of health, aimed at curbing the current resource intensive hospicentric and curative service focus.

Community participation, which is universally acknowledged as a pillar of effective primary health care, underpins the right to health. In this regard, the establishment of equitable partnerships achieves prominence in the Department of Health strategic plan, NDP, and NHI. Other key interventions to implement the NHI are: to improve throughput from training institutions, to address key human resources for health requirements [19], the pooling of resources directed at implementing primary health care interventions, and tracking the effect of health outcomes, which demonstrates accountability. 

### 4.2. Services

Pharmaceutical service provision in South Africa reflects the high level of inequality which characterizes South Africa’s unequal society. On the one hand, private sector hospitals and community pharmacies remain adequately stocked with medicine supplies, provide more efficient service, and operate with optimal coordination; in contrast, public sector hospitals and their pharmacies depict overloaded patient waiting areas, with staff operating with limited resources and under stressful conditions, which are augmented by frequent medicine stockouts [22]. Indeed, only one quarter (26%) of registered pharmacists are employed in the public sector, yet they are responsible for serving the majority (79%) of South Africa’s population [23]. In contrast, pharmacists practicing in private sector community pharmacies, at the primary health care level, operate independently or through corporate chain stores, located in urban areas and offering dispensing services to the 15% minority insured population. Therefore, a patient-centered educational focus would be suited for private sector community pharmacies and hospital level of care. Indeed, patient-centered disease prevention activities relate mainly to community pharmacy practice. Currently, South African pharmacists’ public health role has been categorized into macro- and micro-level pharmacy activities, which reflect population-based and patient-centered disease prevention approaches, respectively [24].

The population focus (operating at the macro level) includes medicine policy regulation, treatment guidelines, essential medicine lists, pharmacoepidemiology, pharmacovigilance, and pharmaceutical services (clinical governance, quality assurance, monitoring and evaluating, drug supply management, and the expanded program on immunization). The individual disease prevention and health promotion focus (operating at the micro level) entails screening and monitoring (for blood pressure, cholesterol, diabetes), education (for smoking cessation, weight management), family planning and immunization, human immunodeficiency virus testing, adherence counselling, and drug misuse awareness. However, the FIP’s needs-based professional educational model does provide room for these aspects to be developed in its vision for interdependence between locally determined needs, socially accountable services, globally connected competencies, and quality assured education [4]. Developing countries could therefore maximize the population health approach to develop a needs-based pharmacy education and practice framework.

However, in addressing the current workforce deficiency and the high patient load in the public sector, along with external influences that the underserved community experience, the availability of adequate pharmacist posts in the public sector remain a challenge. A low pharmacist to population ratio (1:3849), places an additional burden on nursing staff, resulting in suboptimal service provision in significant parts of the public health care system [23]. The government’s drive to increase midlevel workers, such as basic and postbasic pharmacist’s assistants and the introduction of pharmacy technicians, is one of the government’s strategies aimed at addressing the public sector’s pharmacy workforce shortage [25]. Since the latter two cadres’ scope of practice under indirect supervision of the pharmacist, enables dispensing services at PHC clinics, the pharmacist’s responsibilities could then be tailored to population health activities. Following South Africa’s re-engineering of PHC services towards a district-based health system through the National Health Insurance pilot sites, the role of the pharmacist in the district management and special support teams [26] are yet to be clearly defined, as pilot outcomes are finalized [27] However, currently pharmacists conducting population-based health activities are confined to function at the district level of the public health care system, with intermittent contact with pharmacists at facility level.

### 4.3. Competency

In terms of competency, the South African Pharmacy Council (SAPC) is the national regulatory body which is responsible for registration of pharmacy personnel, pharmacies, and the accreditation of nine pharmacy schools. In South Africa, competence standards for entry level pharmacists were first developed [24] to focus on knowledge and skills (Table 1), but behavioral and attitude traits were omitted. Of the seven competence standards (2006), two (6 and 7, Table 1) were directed at promoting community health and participating in research [28]. 

The 2006 competency standards for pharmacists were revised to include behavior and attitude traits depicting alignment with the FIP global competency framework (2012) [4], and contextualized to South African resources, needs, and policies: The NHI, NDP, and National Core Standards [29]. An organized cluster of competencies consisting of six domains for a competency framework suitable for the South African context was developed (Table 2). Of the six domains, one is patient-focused (2), two are population-focused (1, 3) with parts 3.5, 3.6, and possibly 3.7 also being patient-focused; while three domains (4, 5, 6) could overlap between population and individual health categories. Based on the revised competency standards (2017), the public health aspect was expanded, showing promise towards the pharmacist’s role in population health.

### 4.4. Education

South Africa’s undergraduate pharmacy education spans 4 years, thereafter graduates (B.Pharm) complete an internship (1 year), and, after successful completion of a preregistration examination (conducted under the auspices of the South African Pharmacy Council), enroll for mandatory community service (1 year). At entry level, they are recognized as a generalist pharmacist, and seek employment as a registered pharmacist in any sector of the pharmacy profession. Alternatively, a B.Pharm graduate interested in embarking on an academic career, will be required to undertake an academic internship (minimum 2 years), to graduate with a M. Pharm, followed by a PhD (minimum 3 years). A generalist pharmacist specializing in clinical pharmacy (M. Cl Pharmacy) would be recognized as a clinical pharmacist, attains competency to engage in policy decision-making at national and provincial levels, and offers support to a generalist pharmacists, whose primary role is to ensure that strategies (namely antimicrobial resistance stewardship, pharmacovigilance, drug utilization reviews) are operational at the facility level [26]. 

South Africa’s higher education landscape is enduring significant pressure due to rising tensions between university management and students. From the top decision-making structures, the South African Higher Education White paper (2013) states that higher education institutions should develop thinking citizens, who can function effectively, creatively, and ethically as part of a democratic society, and that citizens should have an understanding of their society and be able to participate fully in its political, social, and cultural life [30]. At the bottom end of the higher education institution’s hierarchical ladder are students, whose violent protests (2015 and 2016) expressed their discontentment within and with higher education post-1994 policies and frameworks on equality, equity, transformation and change, institutional cultures, and epistemological traditions which have not changed very much [31]. Described as the most wide-ranging democratic project [32], the students amplified the inclusion of African indigenous epistemologies to contextualize learning through the realities of local communities [33].

Decolonization has specifically been aimed at scientifically based disciplines and professions like pharmacy [34,35,36]. The primary issue with scientific approaches is its exclusive nature, attaching value to empirical knowledge, minimizing the value of indigenous and community knowledge. Biomedicine traditionally operates through sound, scientific, evidence-based principles, espousing that it is the only responsible approach to attaining therapeutic objectives [37]. In essence it would seem difficult for the scientifically trained mind to appreciate the non-standardized and untested (“unscientific nature”) of other approaches such as traditional African health practices [37]. Such science-based hegemony cements the knowledge–power–authority triad which is pervasive in universities [32]. Current higher (or pharmacy) education is training graduates to enter the market place, but falls short on developing their critical and analytical skills that are required to fundamentally change the status quo that pervades a deeply divided society and economy, in an effort to advance African epistemology [31].

A decolonized curriculum would embrace a quest for relevance to the local situation which can be implemented via two approaches. Firstly, new items could be added to the existing curriculum by retaining the scientific world view and adding segments of African knowledge to denote reform and transformation efforts at a superficial level. Secondly, a rethink of how the object of study itself is constituted, and to reconstruct it to bring fundamental change where teaching, research, and service directed to local communities would be achieved through cocreation of knowledge that is fundamentally relevant to the material, historical, and social realities of the communities in which the universities are located [31,38]. Consequently, such a profound change in what is taught and how it is taught will have to be led by a coalition of students, faculty, and the larger public (stakeholders) to find ways to hold institutions accountable for the transformation of themselves and society [31]. In pharmacy education neither curriculum restructuring nor additional resources are required. A reorientation among multiple stakeholders at higher education institutions, the health services, pharmaceutical policy makers, and community based organizations is necessary to engage constructively in optimizing current resources by aligning to the values of social accountability where there is return on investment. A curriculum based on population health could be a starting point to infuse relevance into their activities and evaluate which type of activities maximize return on investment, which benefits the health system and society, and by guiding a constructive and evidence-based dialogue about decolonization.

## 5. Needs-Based Pharmacy Education: A Framework for Population Health 

A needs-based approach underpinned by population health for pharmacy education in South Africa, will require rethinking of some activities which traditionally have neither been part of the accreditation process of the SAPC, nor viewed within the scope of the pharmacy schools. Firstly, there should be involvement of a wide range of internal and external role-players in the development and assessment of the school’s activities, to ensure relevance, quality, and effectiveness. Secondly, this approach will require pharmacy schools to think integratively about their education, service, and research activities to maximize the school’s resource pool, by adding to that of the health system in order to maximize return on investment. Thirdly, a monitoring and evaluation framework is required to track educational outcomes in the health system in defined local populations, which the school services, thereby demonstrating education’s effectiveness to society.

A needs-based framework which includes the three fundamental aspects emanates from the collaborative work of Larkins and colleagues [39]. The researchers developed a logical framework for health education based on the desire to improve equity, relevance, effectiveness, and quality of health education, measured through graduates and their impact on society. Such an evaluation framework was based on three key questions for a school: How does the school work? What do we do? What difference do we make? (Table 3). This framework incorporates activities that might be traditionally implicit in a school’s conceptualization of their educational, research, and service activities, by adding a further step which evaluates the effectiveness of the school’s activities through an evidence-based approach, as opposed to the business as usual output focus of ‘what is being done’.

Larkins and colleagues attest that, firstly, in terms of organization and planning, “an assessment of values, governance and decision-making processes and partnerships with the health sector, community groups and policy makers [as well as] documentation and understanding of the reference population that the school serves, with particular note of underserved groups” is required. Secondly, evidence and assessment of “graduate outcomes (location, discipline and practice), engagement and effect on health services and community outcomes, cost effectiveness and influence with other schools” need to be tracked [39].

The first question contains one of the most significant changes that a population-based approach requires for the pharmacy school’s organization and planning to consider and redefine its stakeholder involvement. The traditional approach of having a likeminded group of discipline-based medicine experts to make program decisions, would consider embracing the inclusion of lay community members, government officials, students, graduates, nongovernmental organizations etc. Such stakeholders will be involved in determining the inputs, the outcomes, and impact of the school’s activities, not only the educational program in isolation. Furthermore, multi-stakeholder involvement would guide how the pharmacy curricula could integrate or complement allopathic and African traditional healing and medicine approaches. Sources of evidence (obtained from documentation) retrieved from the institution for the defined indicators should be assessed and validated among relevant role-players using measurement tools from interviews, focus group discussions, log and audit data, institutional policies, memorandum of understanding, prizes, and research outputs. 

In terms of the second question educational, service, and research activities will need to be aligned and integrated as far as possible. For educational activities, the pharmacists’ public health role [24] and the Clinical Prevention and Population health curriculum could offer some insight into the initial mapping of a population-based curriculum [40], whereby evidence-based practice, preventive services, health systems and health policy, and community practice are outlined. Along with the technical and discipline-based knowledge, a pharmacy curriculum integrating critical skills and advocacy activities to address disparities would enable pharmacists to have the skills, moral commitment and social contract with society [41]. Underlying pharmacy students’ inquiry into the social, economic, environmental, and cultural practices relating to medicine use among communities, reflexivity and critical analysis of how the status quo reinforces current health outcomes which perpetuates privilege and inequality [31], requires constructive engagement with external stakeholders. 

In terms of service activities, current evidence from a systematic review showed that longitudinal placements in underserved and rural areas with a community-based service-learning approach positively impacted student learning and attitudes towards serving in these communities, and a positive impact in terms of population health outcomes [42]. A South African case-control study further attested that rural exposure influences the practice choice among health professionals in a developing country [43]. From the previous evidence-based service activities that have worked for other health professional education, potential service activities for pharmacy students may include the following: Experiential training from the first year, longitudinal placements (1 year or more) in rural or underserved areas, and student recruitment from such areas would be required to address equity. In terms of service activities, attempts should be made to engage with African traditional healers and medicine use practices within local communities to sensitize students to the value of indigenous and community knowledge.

In addition to being integrated into school activities, research activities would entail tying up together evidence retrieved from each of the three questions. In terms of question one, research focusing on a participatory approach based on strengthening of community and practice-based methodologies is required to expand a cocreated evidence base, in which curricular relevance and societal impact could be instituted. Community-based participatory research through collaborative identification of therapeutic sources and formulation of appropriate dosage forms, which could be tested through clinical trials. In addition, the entire school’s research needs to be aligned with priority heath needs, and mechanisms for community feedback on the dissemination and translation of research findings need to be in place to ensure maximal benefit for local communities. Indeed, the inclusion of science engagement, which is the translation of science (knowledge) to communities, is regarded as a compulsory section in research funding proposals of the National Research Foundation of South Africa [44]. For question three, research into longitudinal cohort studies measuring population health indicators in local and participating communities and pharmacy graduate tracking are required to determine which educational programs lead to quality population health outcomes and graduates that fit for purpose, respectively [45]. This would provide the evidence base for educational relevance to examine the return on investment of publicly funded institutional activities. 

## 6. Conclusions

In a resource limited context, needs-based pharmacy education would have more impact if resources were pooled and employed for interventions that would benefit a large part of the population who have been marginalized. To achieve this, pharmacy schools will have to embrace three key mind-shifts, namely: the involvement of numerous external stakeholders to assist with development and assessment of institutional activities; the pooling and integration of school resources of educational, service, and research activities; and the development of a monitoring and evaluation framework that tracks the effectiveness of educational outcomes. The phasing in of this approach cannot mean “business as usual” whereby applying the tick box approach, which was historically part of the reproductive machinery of higher institutions, seemed to be the norm [32]. The population health approach depends on commitment from higher educational institutions to create an evidence base of their impact, in terms of relevance, equity, effectiveness, and quality of their activities to transform higher education, the health system, and society.

## Figures and Tables

**Table 1 pharmacy-07-00116-t001:** Seven competence standards for South African pharmacists (2006) [28].

1.	Organise and control the manufacturing, compounding and packaging of pharmaceutical products.
2.	Organise the procurement, storage and distribution of pharmaceutical materials and products.
3.	Dispense and ensure the optimal use of medicines prescribed to the patient.
4.	Provide pharmacist initiated care to the patient to ensure the optimal use of medicine.
5.	Provide information and education on health care and medicines.
6.	Promote community health and provide related information and advice.
7.	Participate in research to ensure the optimal use of medicines.

**Table 2 pharmacy-07-00116-t002:** Summary of domains and competencies for South African pharmacists (2017) [29].

Domains	Competencies
1. Public Health	1.1 Promotion of health and wellness 1.2 Medicines information 1.3 Professional and health advocacy 1.4 Health economics 1.5 Epidemic and disaster management 1.6 Primary health care 1.7 Pharmacovigilance
2. Safe and rational use of medicines and medical devices	2.1 Patient consultation skills 2.2 Patient counselling skills 2.3 Patient medicine review and management 2.4 Medicines and medical devices safety 2.5 Therapeutic outcome monitoring 2.6 Pharmacist initiated therapy
3. Supply of medicines	3.1 Clinical trials 3.2 Medicine production according to GMP 3.3 Supply chain management 3.4 Formulary development 3.5 Medicine dispensing 3.6 Medicine compounding 3.7 Medicine disposal/destruction
4. Organisation and management skills	4.1 Human resources management 4.2 Financial management 4.3 Facility and infrastructure management 4.4 Quality assurance 4.5 Change management 4.6 Policy development
5. Professional and personal practice	5.1 Patient-centred care 5.2 Professionally practice 5.3 Ethical and legal practice 5.4 Continuing Professional Development 5.5 Leadership skills 5.6 Decision-making skills 5.7 Collaborative practice 5.8 Self management skills 5.9 Communication skills
6. Education, Critical analyses, and Research	6.1 Education and training policy 6.2 Provision of education and training 6.3 Practice embedded education or workplace education 6.4 Gap analysis 6.5 Critical analysis 6.6 Research 6.7 Supervision of other researchers 6.8 Collaborative research

**Table 3 pharmacy-07-00116-t003:** Outline of population health indicators to assess equity in pharmacy education [39].

	Population Health Indicators to Assess Equity
How does the school work? (*Frequently neglected in accreditation process*)	Assessment of the school’s organisation and planning (i.e., values, governance, and decision-making processes. Partnerships with the health sector, community groups, and policy makers. Documentation and assessment of the underserved population which the school serves.
What do we do? (*Corresponds to usual accreditation process*)	Assessment of the students, teachers, curriculum, adult learning approaches, research, service, and resource allocation.
What difference do we make? (*Considered outside the scope of health professional education*)	Assessment of graduate outcomes i.e., location, discipline, and practice. Graduate engagement and effect on health services and community outcomes cost effectiveness and influence with other schools.

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
