# Peer review of "Exploring the Implications of a Needs-Based Pharmacy Education Framework Modelled on Population Health: Perspective from a Developing Country"

_pharmacy, 2019, doi:10.3390/pharmacy7030116_

Reviewer 1 Report

Thank you for your very interesting article covering health inequalities, specifically related to pharmacy education and services, in SA. Although not set out in the traditional IMRaD style, this narrative approach produced a comprehensive articulation of the problems in theis under-develpped country with major issues of inequality of access to health services. The findings and conclusion address the research questions set out. A few minor typos and formatting issues which I'm sure will be picked up at copy-editing stage. I wish teh authors well for dissemenation of their article.

Author Response

Dear Reviewer 1

The authors are gratefully acknowledge the time and effort that was used in reviewing the manuscript.

No changes were required.

Regards

Angeni Bheekie

Reviewer 2 Report

Thanks are expressed to the editor and authors for the opportunity to review this interesting concept paper. The manuscript describes and proposes elements of a needs-based pharmacy education model in South Africa, as an example of a developing country. The foundation of the conceptualized needs-based education model is understandable, though more clarifications and considerations could result in a better developed paper. Specific comments follow.

Background

·         Line 38: International Pharmacy (sic) Federation  (FIP); correct name is International Pharmaceutical Federation

Global pharmacy education guidelines

·         Line 95 notes lack of key aspects on interprofessional teamwork. That’s changing for U.S. health education, e.g., see:  https://healthprofessionsaccreditors.org/wp-content/uploads/2019/02/HPACGuidance02-01-19.pdf

·         Table one, beginning Line 97
- While reference citation 4 provided in title, this is a retyped version of the exact table in the source cited. Editor, for such verbatim copying, is more than a simple citation needed?
- Only difference in words from original source is exclusion of CPD (Continuing Professional Development) in lower right quadrant for Practice Focus.
- As typed, hard to interpret.  “Population focus” spread across two rows in upper left. If table retained, might be easier to read if you moved up “Management Knowledge” at bottom to lead the second half rather than show up at end. In original source, had some shadings to ease reading by focusing the eye in sections.

·         Line 103: Based on FIP, said “Developing countries are therefore required …” Required is not appropriate verb. FIP has no requirement authority though the organization has practice standards, guidelines and related documents.

Needs

·         Lines 112-113 state: “…. predominantly of indigenous (“black”) Africans (80%).”
Admittedly the authors are quite familiar with the culture (vastly more than this reviewer), but my initial reaction was that conflating indigenous with “black” Africans seemed an over simplification of indigenous South Africans.
- Indigenous populations tend to have cultural and migratory underpinnings beyond race. Please provide a reference for this demographic conclusion. If you mean Black South Africans (most of whom might not be classified as indigenous), so state. From my quick scan online, indigenous South Africans are Khoe-San population groups, comprising ~1% of the population.
In addition to clarity needed here, seems important since recommendations regarding inclusion of African indigenous epistemologies listed in lines 215 and 228 under Education section.

·         Line 146, “PHC” abbreviation not defined; also appears later on line 184.

·         Given this reviewer’s comment about Table 1, wonder if Table 2 also just a retype from the source citation though I do not know. Is this full list needed in Table 2, or could the paper summarize win narrative with some examples. I leave that determination to the editor and authors.

·         Quite interesting section conclusion sentence on lines 185-187. If pharmacists conducting population-based health activities are confined to function at the district level of the public health care system, with intermittent contact with pharmacists at facility level, seems the concept paper should also discuss potential need to restructure the system, not just pharmacy education.

Competency

·         Table 3, beginning line 194.
At least #6 and #7 discuss need to promote community health and participate in research, which would buttress some of the suggestions in manuscript section on needs-based pharmacy education, especially some of the recommendations on lines 283-322.

·          Paragraph on lines 195-202: Why is domain 3 considered only to be population-focused? Could parts 3.5, 3.6, and possibly  3.7 also be patient-focused? Paragraph does not mention what is focus of domain 6.

Education

·         For global readers, would be helpful to briefly summarize the current pharmacy education curriculum in South Africa. For example, is it a B.S. degree (4 or 5 years)? Other baccalaureate degree? Master’s or PharmD (if so, how long is curriculum)? I ask because when reading some of the suggested inclusions (starting line 230; then beginning lines 287 and 307, for example), wondered how that would impact the length of the required curriculum.
- Should suggested curricula additions be for all pharmacist education, or could they be for a subset, e.g., who might get a certificate or master’s degree in the area?

·         The need for the conceptualized curriculum would have a stronger basis to advocate with a bit more thought on resources needed for such changes, including time to degree, inclusion of certain subjects, and possible elimination of others. This in backdrop of the population needs (already noted a low pharmacist to population ratio on lines 178-179) and possible regulatory changes needed, such as accreditation standards that were briefly mentioned.

Author Response

Dear Reviewer

The authors gratefully acknowledge the constructive comments to improve the quality of the concept paper. 

Regards

Angeni Bheekie

Reviewer 3 Report

The authors addressed interesting and important topic about allocation resources and responsibility of HEI to meet society needs through curricula, which is not just the same in different countries.

The article could be some kind of opinion paper (and not research article) but needs to be improved.

A/ To persuade readers about importance of a needs-based pharmacy educational approach, authors made some short-cuts, e.g

Lines from 58 on: System based approach to reforming health education

They wrote that the demands of the health care system are determined by population health needs, which is only partially thru. The »system« is determined by many of factors, which have different values in developed or developing countries. This part needs to be elaborated, since the main reason for proposal the needs-based education instead of patient-oriented are (by authors) differences among developed and developing countries. These differences are here neglected.

B/ One of the problematic points of the manuscript are authors’ liberty in modification of cited data without commenting such modifications, for example:

Line 97 and further, Table 1 is bad version of Table from cited document. Authors’ contribution to it resulted in less clarity of given information.

Line 112 and further, Needs: In the original cited source from 2018 (https://www.statssa.gov.za/publications/P0302/P03022018.pdf ), the population of SA is grouped in Black Africans, Coloured, Indian/Asian and White. In the manuscript, the word indigenous is used, while black is put into brackets and apostrophe. Why?

C/ Line 132 and further: Strategic plan (reference 17) is cited several times, but it is inaccessible with given info under reference 17. The cited data from the Strategic plan is difficult (some of them impossible) to find in the Plan, using keywords from the manuscript.

D/ The manuscript deals with educational approach(es) for health care providers on the systemic level but only scares info about educational system in SA is given. The suggested changes/reorientation in competence model is difficult to understand or support without knowledge of the present situation: type(s) and duration of studies for health care providers, especially pharmacists.

Author Response

Dear Reviewer

The authors gratefully acknowledge the constructive comments that were provided to improve the quality of the concept paper.

Details to reviewers response is attached.

Round  2

Reviewer 2 Report

Authors were responsive to reviewer concerns and addressed adequately.

Author Response

Dear Reviewer 2

No further changes were required.

Thank you for your efforts to review the draft manuscript.

Reviewer 3 Report

I read the revised manuscript and the authors' replies. Indeed they did not answer on my comment about Lines from 58 on. It was comment on looks that we differently understand determination of the health system – not the policy makers/government decision about orientation of the system. It would be worth to elaborate this question. Nevertheless, the revised manuscript overcome some weaknesses from the original version.

Author Response

Dear Reviewer 3

Thank you for revisiting the value based health care system for developing and developed countries.

The authors reply is attached 
